# Oscillations of EPR Signals Accompanying Belousov–Zhabotinsky Reaction

Roman Morgunov [1,2,*] and Yoshifumi Tanimoto [3]

1 Institute of Problems of Chemical Physics, 142432 Chernogolovka, Russia
2 Depts of Medical and Biology Physics, I.M. Sechenov First Moscow State Medical University, 119991 Moscow, Russia
3 Graduate School of Advanced Science and Engineering, Hiroshima University, 1-3-1, Kagamiyama, Higashihiroshima 739-8511, Japan; yt1112@hiroshima-u.ac.jp
* Correspondence: morgunov2005@yandex.ru; Tel.: +7-9151382936

**Abstract:** Periodical transformation of ferroin to ferriin is accompanied by changes in magnetic properties of liquids during Belousov–Zhabotinsky (BZ) reaction malonic acid, sodium bromide, sodium bromate, ferroin, and sulfuric acid. Instead of the earlier studied oscillation of microwave conductivity accompanying an oscillating reaction, we propose a flash technique to interrupt the BZ reaction by rapid freezing. Rapid cooling of a solution during chemical oscillations results in a frozen system with a fixed concentration of paramagnetic centers $Fe^{3+}$. EPR spectrum recorded at different stages of the interrupted reaction corresponds to the exact concentration of the ferroin and ferriin components. Following unfreezing unblocks the BZ reaction, and oscillations are still observed. A simulated spectrum allows one to distinguish two groups of $Fe^{3+}$ ions of different symmetries. The obtained results are important to explain the earlier observed effect of inhomogeneous magnetic field on BZ reaction front velocity.

**Keywords:** Belousov–Zhabotinsky reaction; electron spin resonance; crystal field spitting

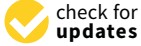



## 1. Introduction

Oscillation of color and physical properties of the solution accompanying the Belousov–Zhabotinsky (BZ) reaction is a classic demonstration of nonlinear dynamics of chemical reaction in open dynamical systems [1,2]. The oscillating transformation of ferriin ↔ ferroin is a known example of a "chemical clock" demonstrating periodical alteration between blue (ferriin) and red (ferroin) color. Oscillatory reactions sensitive to the environment attract the attention of specialists, being very promising for the development of chemical sensors [3]. Additionally, to temperature, humidity and concentration of impurities, BZ oscillations are sensitive to a magnetic field, which affects the propagation of the BZ reaction front [4,5]. The velocity of the BZ reaction front can be changed in an inhomogeneous magnetic field. The effect of the external magnetic field on BZ reaction is a result of magnetic force proportional to the gradient of the magnetic field. This force affects the paramagnetic ingredients of the reaction [4,5]. Our paper is motivated by the necessity to understand what paramagnetic particles involved in oscillating stages of BZ chemical reaction are subjected to magnetic force.

Measurements of dynamics of the reaction by an electron spin resonance spectrometer are complicated due to the finite time of field sweeping close to a period of oscillation. Averaging of the signal of the electron paramagnetic resonance (EPR) during spectrum recording makes constant peaks corresponding to the species not involved in BZ reaction (impurities). Furthermore, high electromagnetic absorption suppresses the Q-factor of the resonant cavity when a liquid sample is installed there. Thus, "in situ" recording of the EPR spectrum in BZ solution is unsuitable for the analysis of paramagnetic particles. Many publications consider periodical changes in the EPR baseline registered during field

sweeping [6,7]. This effect has no relation to the characterization of paramagnetic particles because the analyzed baseline depends on microwave absorption controlled by the electrical conductivity of the sample. Since charged ions are periodically held in intermediate reaction products, one can expect periodical changes in solution conductivity controlling the Q-factor and microwave absorption of a resonant cavity. This effect was accurately described in solid semiconductor samples [8]. Thus, although the EPR spectrometer demonstrates oscillation of the signal during BZ reactions, the results of these experiments are not convenient for the analysis of the time-dependence of paramagnetic ion concentration. We cannot find publications devoted to the analysis of the resonant lines during the BZ reaction.

This paper is aimed at the development of the experimental method for the investigation of the oscillating concentration of paramagnetic ions during the BZ reaction. We propose flash-freezing of the BZ solution to fix periodically changing concentrations. The origin of the magnetic field effect on BZ reaction can be clarified if one considers the obtained information on the concentration of paramagnetic centers in the BZ reaction. We follow the method developed in [9] for cryo-oscillations in the BZ reactions.

## 2. Results

Immediately after mixing the chemicals, the color of the reaction solution periodically changes from blue to red for a few minutes (Figure 1). The red color of the solution corresponds to diamagnetic ferroin-containing $Fe^{2+}$ ions, while the blue one corresponds to paramagnetic ferriin-containing $Fe^{3+}$ ions. After freezing the solution, EPR spectra were recorded at 100 K. Color of the solution at the moment of its freezing was distinguished by eye.

|  (a)  |  (b)  |

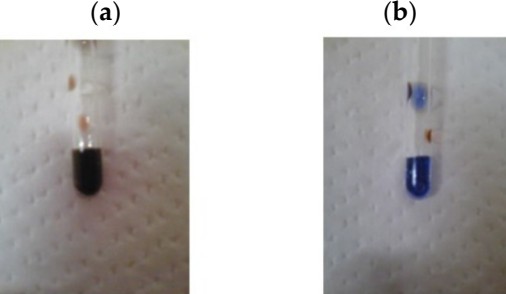

**Figure 1.** Solution with developing Belousov–Zhabotinsky (BZ) reaction at different moments separated by 120 s. Blue (**a**) and red (**b**) color indicates different stages of chemical oscillations corresponding to ferroin (**a**) and ferriin (**b**), respectively.

EPR spectra of the BZ solution contain three signals at $H_{res}$ = 1500 Oe, 2500 Oe, and 3450 Oe at room temperature (lines 1–3 in Figure 2). Correspondending g-factors equal to $4.61 \pm 0.05$ (line 1), $2.63 \pm 0.05$ (line 2), $2.05 \pm 0.05$ (line 3) were calculated by a standard formula $g = h\nu / \mu_B H_{res}$ ($\mu_B$ is Bohr magneton).

Line 1 demonstrates the monotonously increasing intensity I with time (Figure 3). Line widths remain unchanged during the reaction. For this reason, the intensity *I* is proportional to the spin number found by the double integration of the spectrum. Changes in intensity of line 3 slightly increase, but intensity variations are close to the error bar because the results of the calculations depend on the baseline subtraction. Thus, the intensities of lines 1 and 3 are independent of solution color. Many other research works on the $Fe^{3+}$ ions in disordered media reveal EPR spectra containing two lines at g = 2.02 and g = 4.3 (see review [10]). This similarity allows one to suppose that lines 1 and 3 correspond to other groups of paramagnetic ions than those corresponding to line 2.

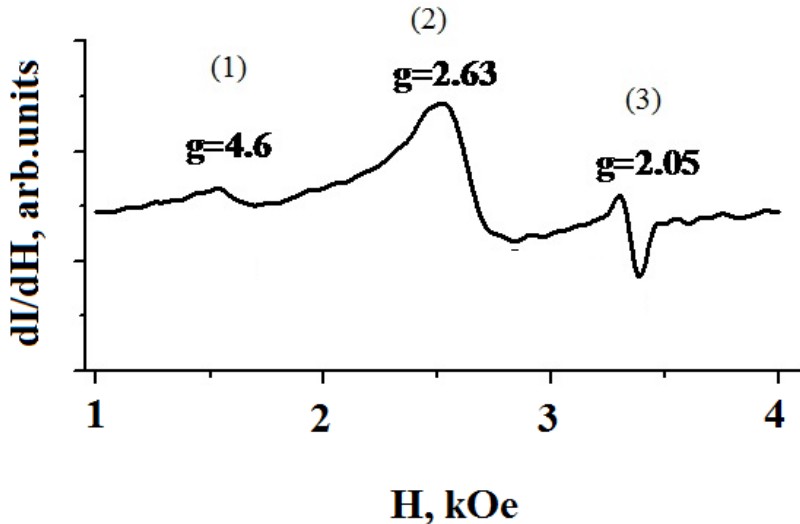

**Figure 2.** Electron paramagnetic resonance (EPR) spectra of ferriin in frozen BZ solution at 100 K.

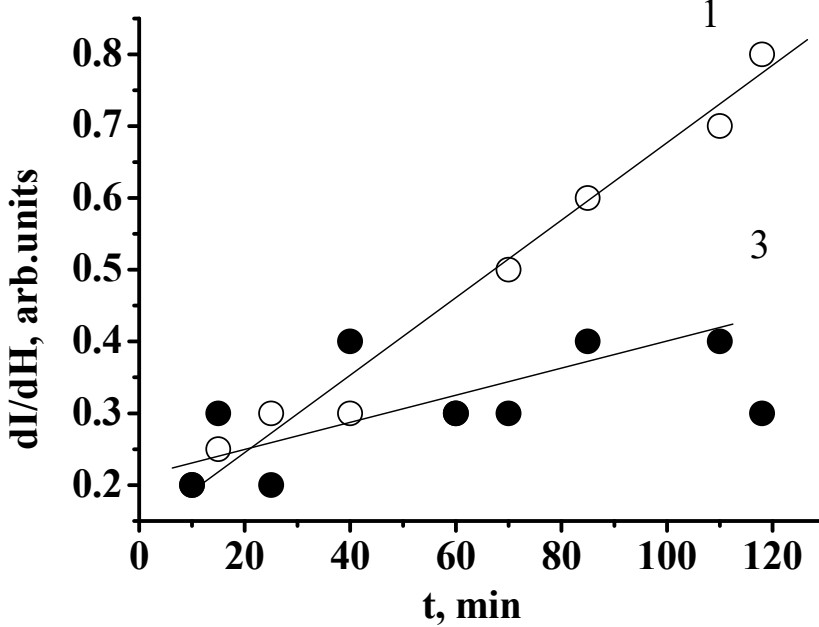

**Figure 3.** Time dependence of intensity I for line 1 and line 3.

The intensity of line 2 at g = 2.84 ± 0.1 oscillates with time (Figure 4). We recorded the spectra centered in 2500 Oe magnetic field in a narrow field range, including line 2, to prevent possible delay and development of possible low-temperature reactions. A low amount of the sample requires a high time constant per one point. The absence of oscillations for lines 1 and 3 makes it reasonable to record line 2 only. EPR spectrum recording requires ~10 min. This time should be as shorter as possible to prevent possible low-temperature processes. Line 2 increases when the solution is blue and decreases when the solution is red. The color of the background in Figure 3 corresponds to the color of the solution at the moment of its freezing. Since line 1 at g = 4.43 ± 0.1 undergoes no oscillation as well as line 2 at g = 2.05 ± 0.05, one can consider these two lines belonging to separated systems. On the contrary, line 3 demonstrates oscillation of its amplitude. For this reason, line 3 corresponds to another type of $Fe^{3+}$ ions with different symmetry of surrounding ligands. An additional small line can be distinguished in Figure 3 on the background of line 2. The amplitude of this small line increases monotonously as the reaction develops, as shown in Figure 4.

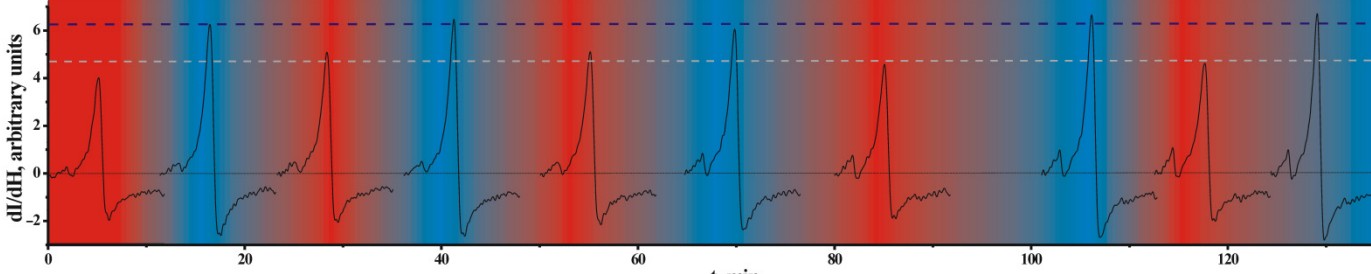

**Figure 4.** Time evolution of EPR line 1 recorded at 100 K in frozen solution freely oscillating during BZ reaction at 295 K. Background colors correspond to the colors of the solution at the moment of its freezing. Horizontal dashed lines correspond to minimal and maximal amplitudes of EPR lines during oscillation.

The amplitude of oscillation for line 2 is about 20% though changes in color look to be correspondent to a major portion of the ions forming the solution color. This fact can be explained by the delay between the appearance of blue color and solution-freezing. Most probably, the reaction continues for 10–20 s after the installation of an EPR tube in the cavity. In this case, 10–20 s is quite enough to decrease the concentration of ferriin in comparison with the maximum one. One cannot exclude the presence of a large amount of the iron ions not involved in the BZ reaction because convection and space oscillations of the reaction products affect a number of Fe ions simultaneously available for EPR detection. Thus, one of three EPR lines recorded in frozen solution oscillates synchronically with its color, while other lines do not oscillate. We can successfully distinguish Fe ions participating in BZ oscillations by flash-freezing technique.

## 3. Discussion

EPR spectra were recorded in the frozen solution. For this reason, we discuss EPR in the frames of the theory of paramagnetic centers in disordered solids (glass or poly-crystals [10]). Since the EPR spectrum of $Fe^{2+}$ ions can be recorded below 20 K with a high-frequency EPR spectrometer, we can exclude the contribution of these ions to the experimental spectrum. Organic radicals usually show a g-factor close to that of the free electron (2.0023) and to g-factor of $BrO_3^{2-}$ complexes included in BZ reaction. Since g-factors of all lines detected in our experiment are far from g = 2.0023 [11], one can conclude that iron phenanthroline complexes $Fe(phen)_3$ with $Fe^{3+}$ ions can contribute to EPR signal, only.

Absorption at g = 4.6 (line 1) is characteristic of isolated $Fe^{3+}$ ions predominantly located in rhombically distorted octahedral or tetrahedral environments. The free trivalent iron ion possesses five unpaired electrons in the 3d-shell ($3d^5$) with zero angular momentum corresponding to the $^6S$ ground state of the ion with spin S = 5/2. In the crystal field, the degeneracy of the high-spin $Fe^{3+}$ state is lifted into three Kramers doublets under the gradient of an internal electric field of the ligand. The remaining degeneracy may be lifted by an external magnetic field resulting in six Zeeman levels. The g -value is expected to be close to the free-ion value of 2.0023. Nevertheless, a g-value higher than 2.0 often occurs [12–14]. Isotropic g = 4.2 value occurs when certain ligand symmetry is present. High g values can be predicted by the spin Hamiltonian [12]:

$$H = \mu_B g H S + D\,[S_z^2 - S(S+1)/3] + E\,(S_x^2 - S_y^2) \tag{1}$$

where $S_x$, $S_y$, $S_z$, are projections of spin on crystalline axes x, y and z, D and E are the second-order crystal field terms with axial and rhombic symmetry. The g value is expected to lie very near the free-electron value of 2.0023 at D = E = 0. Exceeding of D and E over $\mu_B g H$ strongly affects the g value. The values of D and E higher than $\mu_B g H$ result in the two limiting cases, namely, $D \neq 0$, E = 0 and D = 0, $E \neq 0$ when the energy levels in zero magnetic field split into three Kramers doublets $|\pm 1/2\rangle$, $|\pm 3/2\rangle$ and $|\pm 5/2\rangle$.

If $D \neq 0$, $E = 0$, the lowest doublet results in $g = 2.0$ and $6.0$ lines. If $E \neq 0$, $D = 0$, the middle doublet gives an isotropic line at $g = 4.3$–$4.6$. Variation of the g-factor around $g = 4.3$ was explained by the admixture of different $| \pm m_j \rangle$ states caused by the presence of low-symmetry term E $(S_x^2 - S_y^2)$ in the spin Hamiltonian (1) [13]. In [14], the origin of the resonance line of $Fe^{3+}$ ions at $g = 4.3$ was explained by rhombic distortion from cubic symmetry. The electron–hole delocalized among three $t_2$ orbitals mainly populates the $d_{xy}$ state. The shift of the g-factor indicates that the complex is significantly distorted from octahedral geometry due to crystal lattice symmetry or Jahn–Teller distortion of the $t_2^5$ manifold in frozen microcrystals. Thus, resonance line 1 at $g = 4.3$ is due to complexes or crystalline environment where the interaction energy between the surrounding crystal field and the $Fe^{3+}$ ion is higher than the Zeeman energy, and relatively low symmetry takes place.

Most probably, $Fe^{3+}$ ion belonging to ferriin has octahedral ligand environment similar to that known for $Fe(CN)_6^{3-}$ complexes. The ligand field of these complexes is well known and described in the literature [10–13]. The low-spin state of the $d^5$ electronic shell is filled by 5d-electrons on the levels having $t_{2g}$ symmetry. The spin of the $Fe^{3+}$ ion in a low-spin state is $\frac{1}{2}$. EPR spectrum of the low-spin $Fe^{3+}$ complex corresponds to a single line at $g = 2.6$ [15]. The highly covalent bond of an iron ion with organic atoms can explain the strong shift of the line from $g = 2$ value in ferriin as well as in $Fe(CN)_6^{3-}$ complexes [15]. In [16] (see Supporting materials), $g = 2.7$ was found in $Fe(phen)_3^{3+}$ complexes similarly to those participating in the BZ reaction. This value is very close to the $g = 2.8$ value of line 2 found in our experiments.

Finally, line 3 at $g = 2.05$ most probably belongs to high-spin $Fe^{3+}$ ions in an undistorted environment with octahedral symmetry corresponding to $g = 2$. Spin-orbital interaction $\lambda \sim 100$ cm$^{-1}$ and zero-field splitting $\Delta \sim 2.5$–$3 \times 10^4$ cm$^{-1}$ were experimentally determined and theoretically calculated for $Fe^{3+}$ ions in $K_3Fe(CN)_6$ ionic crystals [17]. Orbital triplet splitting is controlled by spin-orbital interaction $\sim \lambda^2 / \Delta \sim 1$ cm$^{-1}$ close to the Zeeman energy. Theoretical calculation of g-factor by spin Hamiltonian (1) results in $g = 2.01$ for $Fe^{3+}$ ion.

Thus, we can conclude on the contribution of the three types of $Fe^{3+}$ ions to the EPR spectra. Different time dependences of the EPR amplitude of these three lines during BZ reaction (Figures 3 and 4) confirm this viewpoint. Two lines (1) and (3) at $g = 4.6$ and 2.05 increase monotonously and simultaneously, is not involved in oscillations. Time dependence of the amplitude of line 2 at $g = 2.63$ is oscillating. Non synchronized time behavior of the EPR lines confirms the presence of a few different types of $Fe(phen)_3^{3+}$ complexes. Rapid cooling of the solution may result in an overcooled nonequilibrium state of the frozen medium containing ferriin molecules in different lattice environment. Different time dependences of the amount of undistorted and distorted $Fe^{3+}$ ions are the most interesting results of our work, indicating intermediate products of BZ reaction in different crystal fields. Complexation of metal ions with macrocyclic ligands may also stabilize unusual oxidation states of metal ions. On the other hand, the variations in kinetic parameters may be associated not only with the possibility of the formation of iron(III) tris-phenanthroline. Ferroin can also be oxidized by anion of hypobromous acid, $BrO^-$, via a one-electron mechanism [18]. Detection and attribution of the intermediate Fe ions in the BZ reaction require future experiments and deeper analysis to distinguish elementary stages of the BZ reaction.

## 4. Materials and Methods

Malonic acid 0.1 M (Merck; reagent grade, >99%), 0.15 M sodium bromide, 0.03 M sodium bromate, 0.004 M ferroin, and 0.34 M sulfuric acid were used. All compounds were of analytical purity grade. All stock solutions were prepared from doubly distilled deionized water. The solution was placed in an EPR quartz holder of 4 mm-diameter, and color oscillations started immediately. The samples were analyzed in a rectangular microwave cavity of a Bruker EPR-300 EPR spectrometer at the Institute for Molecular Sciences, Okazaki, Japan. The microwave magnetic field was of $\nu = 9.567$ GHz frequency

and 10 mW power. The frequency was calibrated using diphenyl picrylhydrazyl (DPPH) reference samples. EPR signal was measured as the first derivative dI/dH of microwave absorption I. The modulation amplitude and frequency were 10 Oe and 100 kHz, respectively. The BZ reaction started, and periodical changes in solution color from red to blue were observed. The approximate period of changes in color was 5 min at 295 K. When a blue color was observed, the EPR capsule was immediately placed into the resonant cavity cooled to 100 K. Continuous flow of liquid helium was provided by pumping through the resonant cavity. The solution froze during few seconds after installation of the 4 mm-thin walls Suprasil quartz tube into the EPR cavity. EPR spectra were recorded after rapid-freezing (~10 s) and thermal stabilization (1–2 min) of the solution. During spectra recording, the reaction was fully blocked, and no changes were observed at 100 K temperature. After the spectrum recording, the sample was removed from the cavity, and its temperature was increased up to 295 K to continue the BZ reaction cycling. When the blue color turned red, the EPR capsule was mounted to the EPR cavity again. Thus, the reaction was frozen every half cycle, and the EPR spectrum was recorded. Sample heating was provided by removing it from the cavity and exposing it to room temperature atmosphere. Storage of the sample for 1 min at room temperature was enough to unlock the ZB reaction due to the thin walls of the tube and the small amount of liquid. Since the average temperature of the solution was lower than in ambient conditions, one can expect a deceleration of the BZ reaction. We did not analyze the effect of temperature on the oscillation period. The magnetic field of the EPR spectrometer was highly homogeneous. No field effect on the kinetics of BZ reaction was possible to find in the absence of magnetic field gradient [8].

## 5. Conclusions

The EPR spectra of the ferriin complexes were recorded in the frozen solution at subsequent oscillating stages of the BZ reaction. The oscillation of $Fe^{3+}$ EPR signal synchronized with color change was observed in the ferroin–ferriin dynamic system. Analysis of the spectra indicates the presence of the two groups of $Fe^{3+}$ ions of different symmetry contributing to the spectra. The amplitude of the EPR spectrum of ferriin complexes with perfect octahedral symmetry is time-independent. Distorted ferriin complexes involved in periodical oscillations of the chemical composition demonstrate oscillating time dependence of the correspondent EPR line. The effect of inhomogeneous magnetic field on BZ reaction observed earlier can be explained by forces acting on the part of $Fe^{3+}$ ions participating in oscillations.

**Author Contributions:** Conceptualization, Y.T.; methodology, Y.T.; validation, R.M., Y.T.; formal analysis, R.M.; investigation, Y.T. and R.M.; resources, Y.T; writing—original draft preparation, R.M.; writing—review and editing, R.M.; visualization, R.M.; supervision, Y.T.; funding acquisition, R.M. All authors have read and agreed to the published version of the manuscript.

**Funding:** R.M. is supported by the Ministry of Education and Science of Russian Federation (Agreement No. 14.W03.31.0001—Institute of Problems of Chemical Physics of the Russian Academy of Sciences, Chernogolovka). The authors are thankful to Institute for Molecular Sciences, Okazaki, Japan, for EPR spectroscopy.

**Conflicts of Interest:** The authors declare no conflict of interest. The funders had no role in the design of the study; in the collection, analyses, or interpretation of data; in the writing of the manuscript, or in the decision to publish the results.

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
