# Peer review of "Oscillations of EPR Signals Accompanying Belousov–Zhabotinsky Reaction"

_magnetochemistry, doi:10.3390/magnetochemistry7010002_

Round 1

Reviewer 1 Report

The oscillating Belousov-Zhabotinsky (BZ) reaction is an endless source of fascinating physico-chemical problems, as exemplified also in the present manuscript, which describes a new approach to the study of the BZ reaction dynamics by means of electron paramagmetic resonance spectroscopy (ESR).

In particular, the focus here is on investigating the dynamic changes in concentration of paramagnetic species during the course of the BZ reaction. Flash-freezing of the reaction mixture enables to quench it for the time required for the acquisition of an EPR spectrum, revealing an instant picture of the paramagnetic species present. The results can shine light on the effect of an external magnetic field on the dynamics of the BZ reaction.

The work is interesting, but it needs to be properly amended before it can be published.

First, the language needs to be thoroughly revised and professionally edited.

Figure 1 could be improved and the chemical structures rewritten in a more correct way.

Line 85: “Most probably this line corresponds to Fe+ paramagnetic centers.”: please explain. +1 is not a typical oxidation state for iron, so if that’s a typo, correct it; otherwise, a detailed explanation is needed.

Lines 86-90: the authors refer to the ESR data for iron centers in metal halogenide crystals in solid state; however, how close these systems can be compared to the case of an organic ligand-based complex in solution (as for ferroin)? Please discuss.

Line 110: please give suitable reference(s).

Line 113: “Simulation of the spectra by Symphonia software”: please give enough details about the software and its operation in the Materials and Methods section.

Lines 134-135: “This stage provides a selection of the complexes with different symmetry.”: please explain the meaning of this expression.

Materials and Methods section: please disclose the purity and source(s) of the chemicals used. How the cooled samples were thawed to room temperature? How much time was required to do so? How the cooling of the cavity was obtained?

Did the authors observed significant changes in the oscillation period for a sample subjected to periodic freeze-thaw cycles and a control one? Please comment!                                                                            

The references section needs to be amended.

Author Response

We are grateful to the referee for his comments. We introduced following changes, marked in text by color. File of rewritten manuscript is attached

  • First, the language needs to be thoroughly revised and professionally edited.

English was improved.

  • Figure 1 could be improved and the chemical structures rewritten in a more correct way.

Chemical structure was improved in Figure1.

  • Line 85: “Most probably this line corresponds to Fe+ paramagnetic centers.”: please explain. +1 is not a typical oxidation state for iron, so if that’s a typo, correct it; otherwise, a detailed explanation is needed.

We are sorry for multiple mistakes. Of course Fe3+ ions everywhere implied in manuscript. Discusion and results sections were fully rewritten.

  • Lines 86-90: the authors refer to the ESR data for iron centers in metal halogenide crystals in solid state; however, how close these systems can be compared to the case of an organic ligand-based complex in solution (as for ferroin)? Please discuss.

We agree with referee. We compared spectrum with the Fe(CN)63+. Also we included comparison with EPR in disordered solids, because we measure EPR in solid sample at low temperature 100 K. Correspondent text was installed in manuscript. Unfortunately we didn’t find EPR in solution, convenient for comparison.

  • Line 110: please give suitable reference(s).

Reference list was improved.

  • Line 113: “Simulation of the spectra by Symphonia software”: please give enough details about the software and its operation in the Materials and Methods section.

We deleted simulation because Bruker software is not available now. Instead of that,  we included discussion of g-factors.

  • Lines 134-135: “This stage provides a selection of the complexes with different symmetry.”: please explain the meaning of this expression.

Explanation concerning microcrystalls of different symmetry was added in the text.

  • Materials and Methods section: please disclose the purity and source(s) of the chemicals used. How the cooled samples were thawed to room temperature? How much time was required to do so? How the cooling of the cavity was obtained?

Purity of chemicals and process of cooling and heating were described.

  • Did the authors observed significant changes in the oscillation period for a sample subjected to periodic freeze-thaw cycles and a control one? Please comment!

We have no accurate measurement of the period in reference sample, but increase of the period was obviously found due to delay of the reaction in solid state.

  • The references section needs to be amended.

Reference section was improved starting from ref. [10].

Reviewer 2 Report

The authors present an epr study of  the Belousov Zhabotinskii reaction in a ferroin –ferrin system. As apparently in situ epr measurements at room temrperature were without any results, they applied a rapid freezing technique to analyze the paramagnetic species at the blue and red phases of the reaction. The approach is interesting and the results would in principal warrant publications.

The manuscript suffers from various deficiencies, some of them are surprising to an extend  that questions of the scientific background of the authors arise.

*The epr results are shown in one single spectrum (fig.1) ; it shows three components labeled by the authors (1), (2) and (3), which are characterized by three g-values of 4.06, 2.63 and 2.05.  They attribute them to two Fe3+complexes with octahedral (1,3) and cubic , octahedrally distorted point symmetry. This assignment requires more scientifique justification to be acceptable. I donot think that principal g-values of ga=4.6 and gb=2.05 are compatible with octahedral symmetry. Further, I wonder how a single g-value of g=2.63 allows them to attribute the spectrum to Fe3+and deduce the point symmetry.

Further :

  • line 84 :the authors discuss the observed g-value of 4.43, which is different from the value given in fig.1. So which is the correct one ?
  • line 86 : Fe3+centers with an electronic d7shell ? this would be an Fe1+state ! However the authors attributed the low field line to before to Fe3+
  • line 95 : large line at g=2.84 corresponds to oscillations of the Fe2+concentration ; just above it was ascribed to Fe3+. In the fig1 the g-value is given as g=2.65 !
  • line 112 : spin of Fe3+in low spin state is S+1/2. Which is the parameter used in symphonia simulation : S=1/2 or S=5/2 ?
  • Figure 2 : I cannot recognize the figure 2 epr spectrum in figure 1. Why is the small component at g=4.43 not present in Figure 1 ?
  • Whereas the color change blue/red is complete, only an intensity change of <20% is shown for the epr spectrum (2). How do the authors explain this ?

In conclusion : the experimental results presented are at best incomplete and their interpretation is not convincing. Many statements are contradictory and the scientific quality of the assignments of the epr spectra  is low. I cannot propose the publication of this manuscript

Author Response

 Answer to referee 2.

We are grateful to the referee for his/her comments. We introduced following changes, marked in text by color.

  • The epr results are shown in one single spectrum (fig.1) ; it shows three components labeled by the authors (1), (2) and (3), which are characterized by three g-values of 4.06, 2.63 and 2.05.  They attribute them to two Fe3+complexes with octahedral (1,3) and cubic , octahedrally distorted point symmetry. This assignment requires more scientifique justification to be acceptable. I donot think that principal g-values of ga=4.6 and gb=2.05 are compatible with octahedral symmetry. Further, I wonder how a single g-value of g=2.63 allows them to attribute the spectrum to Fe3+and deduce the point symmetry.

 We agree with referee and sorry for terrible confusion. Of course all lines correspond to Fe3+ ions surrounded in different environments. We fully reconstructed discussion section, step by step discussing every line. Comparison with Fe3+ systems known in literature is included.

  • line 84 :the authors discuss the observed g-value of 4.43, which is different from the value given in fig.1. So which is the correct one ?

We replaced g-value to correct one.

  • line 86 : Fe3+centers with an electronic d7shell ? this would be an Fe1+state ! However the authors attributed the low field line to before to Fe3+

No Fe+ ions are presented in our work. This is sudden misprint.

  • line 95 : large line at g=2.84 corresponds to oscillations of the Fe2+concentration ; just above it was ascribed to Fe3+. In the fig1 the g-value is given as g=2.65 !

Yes, this is mistake. We installed correct value.

  • line 112 : spin of Fe3+in low spin state is S+1/2. Which is the parameter used in symphonia simulation : S=1/2 or S=5/2 ?

We deleted Symphonia simulation, because it is not available in Bruker website. Instead of that we included comparison with the g-values of Fe3+ known in the literature for different symmetry of environment.

  • Figure 2 : I cannot recognize the figure 2 epr spectrum in figure 1. Why is the small component at g=4.43 not present in Figure 1 ?

We recorded oscillating line only in narrow field range to decrease recording time and prevent possible low temperature processes disturbing BZ reaction. Oscillating line seems to be more interesting than other lines.

  • Whereas the color change blue/red is complete, only intensity change of <20% is shown for the epr spectrum (2). How do the authors explain this?

We proposed an explanation. Installation of the sample to the cavity and start of the spectra recording is later, than color registered. Thus, this delay can decrease number of Fe3+ ions.

Round 2

Reviewer 1 Report

The manuscript has been clearly improved for what concerns the scientific issues. However, the language issues remain, and I strongly advise the authors to seek the help of a native English speaker. The article is of good quality, and it would be a pity to let poor language diminish its values.

I spotted a couple of chemical problems that need to be corrected:

  • Line 122: “BrO3 and BrO32- complexes”: “BrO3” does not exist, while BrO3- does (bromate anion); “BrO32-“ does not exist either. Please correct.
  • Lines 178-181: a “BrO” species is mentioned, which is not known to chemical literature. I suspect the authors meant the anion of hyprobromous acid, BrO-, but it could also be a radical – but then it should be properly labelled. Please correct.

Author Response

Authors are thankful to referee for very useful comments allowing us to improve manuscript. We introduced to the text following changes.

  • English was checked and improved.

  • Line 122: “BrO3and BrO32- complexes”: “BrO3” does not exist, while BrO3- does (bromate anion); “BrO32-“ does not exist either. Please correct.

Phrase was corrected. Just BrO3- was mentioned. The others were deleted.

  • Lines 178-181: a “BrO” species is mentioned, which is not known to chemical literature. I suspect the authors meant the anion of hyprobromous acid, BrO-, but it could also be a radical – but then it should be properly labelled. Please correct.

Anion of hyprobromous acid, BrO- was discussed only. It was labeled.

Reviewer 2 Report

The revised manuscript is even more confused than the first version and does not meet the standards required for publication in this journal

Author Response

Authors are thankful to referee for very useful comments allowing us to improve manuscript. We introduced to the text following changes.

  • English was checked and improved.

  • Phrase on “BrO3and BrO32- complexes”was corrected. Just BrO3- was mentioned. The others were deleted.

  • Lines 178-181: Anion of hyprobromous acid, BrO- was discussed only. It was labeled.
